# 2D In-Plane CuS/Bi_2_WO_6_ p-n Heterostructures with Promoted Visible-Light-Driven Photo-Fenton Degradation Performance

**DOI:** 10.3390/nano9081151

**Published:** 2019-08-11

**Authors:** Li Guo, Kailai Zhang, Xuanxuan Han, Qiang Zhao, Danjun Wang, Feng Fu

**Affiliations:** 1College of Chemistry and Chemical Engineering, Shaanxi Key Laboratory of Chemical Reaction Engineering, Yan’an University, Yanan 716000, China; 2State Key Laboratory of Organic-Inorganic Composites Beijing Key Laboratory of Electrochemical Process and Technology for Materials, Beijing University of Chemical Technology, Beijing 100029, China

**Keywords:** Bi_2_WO_6_ nanosheets, photo-Fenton, degradation, visible light irradiation, 2D in-plane heterostructures

## Abstract

Photo-Fenton degradation of pollutants in wastewater is an ideal choice for large scale practical applications. Herein, two-dimensional (2D) in-plane CuS/Bi_2_WO_6_ p-n heterostructures have been successfully constructed by an in situ assembly strategy and characterized using XRD, XPS, SEM/TEM, EDX, UV-Vis-DRS, PL, TR-PL, ESR, and VB-XPS techniques. The XPS and the TEM results confirm the formation of CuS/Bi_2_WO_6_ heterostructures. The as-constructed CuS/Bi_2_WO_6_ showed excellent absorption in visible region and superior charge carrier separation efficiency due to the formation of a type-II heterojunctions. Under visible light irradiation, 0.1% CuS/Bi_2_WO_6_ heterostructure exhibited the best photo-Fenton-like catalytic performance. The degradation efficiency of Rhodamine B (RhB, 20 mg·L^−1^) can reach nearly 100% within 25 min, the apparent rate constant (k_app_/min^−1^) is approximately 40.06 and 3.87 times higher than that of pure CuS and Bi_2_WO_6_, respectively. The degradation efficiency of tetracycline hydrochloride (TC-HCl, 40mg·L^−1^) can reach 73% in 50 min by employing 0.1% CuS/Bi_2_WO_6_ heterostructure as a photo-Fenton-like catalyst. The promoted photo-Fenton catalytic activity of CuS/Bi_2_WO_6_ p-n heterostructures is partly ascribed to its low carriers recombination rate. Importantly, CuS in CuS/Bi_2_WO_6_ heterostructures is conducive to the formation of heterogeneous photo-Fenton catalytic system, in which Bi_2_WO_6_ provides a strong reaction site for CuS to avoid the loss of Cu^2+^ in Fenton reaction, resulting in its excellent stability and reusability. The possible photo-Fenton-like catalytic degradation mechanism of RhB and TC-HCl was also elucidated on the basis of energy band structure analysis and radical scavenger experiments. The present study provides strong evidence for CuS/Bi_2_WO_6_ heterostructures to be used as promising candidates for photo-Fenton treatment of organic pollutants.

## 1. Introduction

In recent years, with the continuous development of modern economic society, the problem of water pollution is becoming more and more serious [1,2,3]. Especially organic dye and antibiotic wastewater discharged from factories have attracted people’s attention [4,5,6,7]. The complex structure of the dye and antibiotic molecules containing amino groups, carboxyl groups, and benzene rings, lead to toxicity (teratogenic, carcinogenic, and mutagenic) and poor biodegradability [1,2,3,4,5,6,7]. If the dye and antibiotic-containing wastewaters are directly discharged into the freshwater without treatment, the human health and ecosystem will be threatened. Therefore, finding a simple, efficient, and low-cost water pollution remediation method has attracted the attention of researchers.

Among them, advanced oxidation processes (AOPs) have been regarded as promising methods for removing organic pollutants from wastewater, especially the photo-Fenton technology combined with the advantages of photocatalysis and Fenton catalysis, and have been greatly developed [8,9,10,11,12,13]. It is well known that the photo-Fenton technology is based on the enhanced electron transfer between H_2_O_2_ and iron, copper-based catalysts under light irradiation, and then cause high active free radicals such as ·OH to remove organic pollutants [14,15,16,17,18]. Moreover, the heterogeneous Fenton catalyst because of the low metal ion leaching amount, the wide pH application value, and excellent cycle performance has become a research hot spot, and has shown good application prospects [14,15,16,17,18,19,20].

Bi-based semiconductors such as Bi_2_WO_6_ have attracted much attention in the application of photocatalytic wastewater treatment, due to the suitable band gap energy, low cost, and better catalytic performance [21,22,23,24]. It has also been confirmed by combining an additional photocatalyst having a transitional metal oxidation state with Bi-based semiconductors to form heterogeneous photo-Fenton catalysts for pollutants treatment [25,26,27]. As one of the narrow-band gap Cu-based semiconductors with the advantages of being non-toxic, low-cost, and easy to access, CuS has been extensively studied in combination with other photocatalysts for photocatalytic or photo-Fenton degradation of organic pollutants [28,29,30,31]. For example, Gao et al. [28] reported that the introduction of CuS into TiO_2_ by cysteine-assisted in situ synthesis is helpful for rapid transfer and separation of the carriers, thus improving photocatalytic activity. Cai et al. [29] used an in situ synthesis technique to prepare g-C_3_N_4_/CuS p-n heterostructured photocatalyst. The presence of CuS in heterostructure improves the optical absorption and also efficiently facilitates the separation of photo-generated electron–hole pairs. In addition, Bhoi et al. [30] reported that BiFeO_3_ was modified by CuS nanorods through a two-step process to generate a type-II heterostructure, which can enhance the photo-Fenton catalytic activity of BiFeO_3_, resulting from enhanced visible light absorption, greater charge carrier separation, and improved charge mobility. A similar phenomenon has also been observed on CuS/Bi_2_W_2_O_9_ heterojunctions by Bhoi’s group [31]. Under the action of photogenerated electrons, the circulation of Cu^+^ and Cu^2+^ on the surface of photocatalyst can significantly promote the photo-Fenton process, which can produce ·OH more effectively than the traditional Fenton reaction, and ultimately improve the photo-Fenton performance [32].

In this paper, we synthesized CuS/Bi_2_WO_6_ heterostructure by a simple two-step hydrothermal method. The activity results show that the heterostructures can greatly promote photo-generated electron–hole separation, and the photo-Fenton catalytic activity of degradation of Rhodamine B (RhB) and tetracycline hydrochloride (TC-HCl) is significantly improved. In addition, the composition of the heterogeneous Fenton reaction system can significantly improve the recyclability of the catalyst. Moreover, we have investigated the effects of H_2_O_2_ concentration, catalyst dosage, and initial pH value on the photo-Fenton activity, and detailed the possible mechanism of photo-Fenton catalytic reaction.

## 2. Experimental Section

### 2.1. Chemicals

Sodium hydroxide (NaOH, 96.0%) and cetyltrimethylammonium bromide (CTAB, C_1__9_H_42_BrN, 99.0%) were purchased from Tianjin Kemiou Chemical Reagent Co., Ltd. (Tianjin, China). Sodium tungstate dihydrate (Na_2_WO_4_·2H_2_O, 99.5%), bismuth nitrate pentahydrate (Bi(NO_3_)_3_∙5H_2_O, 99%), and absolute alcohol (C_2_H_5_OH, 99.7%) were purchased from Tianjin Zhiyuan Chemical Reagent Co., Ltd. (Tianjin, China). Hydrochloric acid (HCl, 36%), nitric acid (HNO_3_, 63%) and ammonia solution (NH_3_·H_2_O, 28%) were purchased form Sichuan Xilong chemical Co., Ltd. (Sichuan, China). Thiourea (CH_4_N_2_S, 99.0%) was purchased from Sinopharm Chemical Reagent Co., Ltd (Shanghai, China). Copper nitrate trihydrate (Cu(NO_3_)_2_·3H_2_O, 99.0%) was bought from Tianjin Yongsheng Fine Chemical Co., Ltd. (Tianjin, China). The deionized water was used as a solvent. All the reagents were of analytical grade and were used without any further purification.

### 2.2. Sample Preparation

#### 2.2.1. Preparation of Bi_2_WO_6_ Nanosheets

The Bi_2_WO_6_ nanosheets were synthesized by a modified hydrothermal method in accordance with previous report [33]. First, 1.94 g of Bi(NO_3_)_3_·5H_2_O was dissolved into 5 mL of 4 mol·L^−1^ HNO_3_ aqueous solution. To this solution, 0.02 g of cetyltrimethylammonium bromide (CTAB) and 0.60 g of Na_2_WO_4_·2H_2_O in 50 mL of deionized water was then slowly added dropwise to the above solution under magnetic stirring, and the solution of pH is adjusted to neutral with 1:1 concentrated ammonia water (NH_3_·H_2_O), and finally stirred vigorously for 1 h to obtain a white precipitate. The whole mixture was transferred to a 100 mL Teflon-lined stainless steel autoclave and treated hydrothermally at 180 °C for 14 h. After cooling to room temperature, it was washed three times with deionized water and absolute ethanol, dried at 70 °C for 8 h, and calcined at 300 °C for 3 h in a muffle furnace to remove the surface residue.

#### 2.2.2. Preparation of CuS/Bi_2_WO_6_ Heterostructure

The CuS/Bi_2_WO_6_ composites were synthesized by a hydrothermal method according to the previous literature with minor revision [31]. In brief, (0.0024 g, 0.0315 mmol) of thiourea was dissolved in 75 mL water, then 1 g of as-prepared Bi_2_WO_6_ was added under ultrasonication for 10 min. Subsequently, (0.0025 g, 0.0103 mmol) of Cu(NO_3_)_2_·3H_2_O was added with another ultrasonication for 10 min. After stirring for 1 h, the resulting suspension were placed in a 100 mL stainless steel autoclave at 150 °C for 24 h. Finally, when the oven was naturally cooled to room temperature, the obtained products were separated by centrifugation and washed 3 times with deionized water and absolute ethanol, and dried at 70 °C in vacuum overnight to obtain 0.1% CuS/Bi_2_WO_6_ composite (Herein, 0.1% represents the weight ratio of CuS to Bi_2_WO_6_). Following the similar procedure, a series of X% CuS/Bi_2_WO_6_ (X = 0.02%, 0.5%, 1% and 2%) heterostructures were prepared using the same procedure, only the amount of Cu(NO_3_)_2_·3H_2_O and thiourea was changed. For comparison, pure CuS was also prepared by the same procedure in the absence of Bi_2_WO_6_.

### 2.3. Characterization

The XRD patterns of samples were recorded by a Shimadzu XRD-7000 using a Ni filtered Cu Kα (λ = 0.15418 nm) as X-ray source. The XRD data was obtained in 2θ range of 10–80° at a scan rate of 8°·min^−1^. The X-ray photoelectron spectra (XPS) patterns of samples were measured on a PHI-5400 (Physical Electronics PHI, MN, America ) 250 xi system with Al Kα X-rays as the excitation source. The morphology and microscopic size of the samples were carried out by a JSM-6700F (Japan electronics) and a (JEM-2100) (Japan electronics). Energy disperse X-ray (EDX) analysis was performed on a field emission scanning electron microscope (JSM-7610F). The optical properties of the samples were detected on a UV-2550 spectrophotometer (Japan Shimadzu, Kyoto, Japan) within 200 nm to 800 nm, using BaSO4 as a reflectance standard. The photoluminescence (PL) spectra of samples were conducted on a F-4600 spectrophotometer (Hitachi, Japan). Time-resolved photoluminescence (TR-PL) spectra were conducted on a FLS920 fluorescence spectrometer (Edinburgh Analytical Instruments, Edinburgh, UK). The electron spin resonance (ESR) spectra were obtained on a JES-FA300 model spectrometer (ESR, Japan JEOL, Tokyo, Japan) under visible light irradiation (λ ≥ 420 nm). The magnetic central field intensity, scanning width, scanning time, and microwave power are 324.006 mT, 10.00 mT, 60 s, and 0.99800 mW, respectively. The spectra is 4096 points. In a typical experiment for ·OH detection, 10 mg of photocatalyst was dispersed in the mixture solution of 20 mL of ultrapure water and 50 μL of H_2_O_2_ under ultrasonic for 30 min. Then, 10 μL of mixture was sampled and 10μL of 5,5-dimethyl-1-pyrroline-N-oxide (DMPO) (10 ppm) was added with ultrasonic dispersion for 5 min. Thereafter, the obtained mixtures were sampled by a capillary tube and placed into a quartz ESR tube then subjected to test. The signals were collected after visible light irradiation for different time. Following a similar process—just replacing ultrapure water with methanol—·O_2_^−^ was also detected.

### 2.4. Photocatalytic and Photo-Fenton Catalytic Activity Measurement

The prepared samples were used as photo and photo-Fenton like catalysts for the degradation of Rhodamine B (RhB, 20 mg·L^−1^) and tetracycline hydrochloride (TC-HCl, 40 mg·L^−1^). The photocatalytic degradation experiments were carried in photocatalytic reactor (Xujiang Mechanical and Electrical Factory, Nanjing, China, XPA-7). A 300 W metal halide lamp equipped with a 420 nm cut-off filter was used to simulate visible light source. The circulating cooling water was introduced to avoid liquid volatilization and keep the reaction temperature at 25 °C through the temperature control system. Moreover, 0.02 g of the catalyst was added to a 50 mL capacity reactor containing 20 mL of simulated contaminants aqueous solution adjusting with the determined pH (with NaOH or HCl aqueous solution), and then stirred under dark conditions for 2 h to ensure adsorption/desorption equilibrium, then a certain volume of H_2_O_2_ (30%) was added when the light source was turned on. At a certain interval, 2 mL of the supernatant was removed for centrifugation, and then the maximum characteristic absorption peaks of RhB and TC-HCl at 554 nm and 357 nm were measured by UV-Vis spectroscopy (Shimadzu UV-2550), respectively.

## 3. Results and Discussion

### 3.1. Structure, Composition, and Morphology of Samples

The XRD patterns of as-synthesized Bi_2_WO_6_, CuS, and CuS/Bi_2_WO_6_ samples are shown in Figure 1. For the Bi_2_WO_6_ sample, all the diffraction peaks can be assigned to the orthorhombic phase Bi_2_WO_6_ (JCPDS No. 39-0256) [34]. Besides this, the peaks observed for pure CuS are assigned to hexagonal covellite phase (JCPDS No.06-0464) [31]. For the 0.1% CuS/Bi_2_WO_6_ heterostructure, the characteristic peaks of Bi_2_WO_6_ are still retained, but there no characteristic peaks of CuS species and other phases appear in the XRD pattern, owing to the low loading amount of CuS.

To ulteriorly investigate the surface composition and chemical state information of the CuS/Bi_2_WO_6_ heterostructure, XPS measurement was subsequently performed, and the binding energy of C 1s was calibrated at 284.6 eV. The surface survey spectrum (Figure 2a) showed the existence of Bi, W, and O elements in both the Bi_2_WO_6_ and 0.1% CuS/Bi_2_WO_6_ heterostructure, and exemplified only the existence of Cu, S, and C elements in the CuS, in which the binding energy of C 1s is ascribed to the adventitious carbon from the XPS instrument itself. The high-resolution spectra of Bi 4f, W 4f, Cu 2p, and S 2p are shown in Figure 2b–e, respectively. Figure 2b shows two peaks of 0.1% CuS/Bi_2_WO_6_ heterostructure are observed at around 164.57 and 159.25 eV, corresponding to the Bi 4f_5/2_ and Bi 4f_7/2_ energy states of Bi^3+^ [35]. The strong peaks at 37.48 and 35.49 eV are assigned to the W 4f_5/2_ and W 4f_7/2_ for W^6+^ (Figure 2c) [36]. The corresponding Cu 2p peaks are located at 952.42 and 932.02 eV, and are designated Cu^2+^ 2p_1/2_ and Cu^2+^ 2p_3/2_ (Figure 2d) [30,31,37]. It is important to note that the binding energies of Bi 4f, W 4f, and Cu 2p in the 0.1% CuS/Bi_2_WO_6_ heterostructure all exhibited slight shift to the binding energy compared with that of Bi_2_WO_6_ or CuS (Figure 2b–d) due to the interaction between CuS and Bi_2_WO_6_ nanosheets. As for S 2p, it can be seen that the standard peaks of CuS at 163.4 eV and 162.4 eV for S 2p_1/2_ and S 2p_3/2_ were observed, which are consistent with the reported binding energies in metal sulfide. (Figure 2e) [38]. On the contrary, two different peaks are unexpectedly similar to the binding energies of Bi^3+^ at 159.10 and 164.30 eV in the 0.1% CuS/Bi_2_WO_6_ heterostructure (Figure 2e). It may be because the binding energy of S^2−^ is very similar to that of Bi^3+^ [39]. The XRD and XPS results implied that 0.1% CuS/Bi_2_WO_6_ heterostructure was authentically obtained.

The morphology and size of the as-synthesized Bi_2_WO_6_, CuS, and 0.1% CuS/Bi_2_WO_6_ heterostructure were observed by SEM and TEM images. Figure 3a,b depict the nanosheet structures of Bi_2_WO_6_ with the average thickness of about 20 nm. Figure 3c,d show the SEM image of CuS sample presenting spherical-shape morphology assembled by uniform nanosheets with thickness of approximately 50 nm. As can be seen from Figure 3e,f, the CuS nanosheets were well in-plate assembled on the Bi_2_WO_6_ nanosheets, and the tight interfaces between Bi_2_WO_6_ and CuS further confirmed the formation of CuS/Bi_2_WO_6_ heterojunctions. Moreover, the energy-dispersive X-ray analysis of 0.1% CuS/Bi_2_WO_6_ heterostructure was shown in Figure 3g–l, which confirms the uniform distribution of Bi, W, O, Cu, and S species over the 0.1% CuS/Bi_2_WO_6_ heterostructure.

Figure 4a shows the in situ assembly strategy of 2D/2D CuS/Bi_2_WO_6_ heterostructures, which depicts the formation of CuS nanosheets onto Bi_2_WO_6_ nanosheets for assembling in-plane heterostructures. Figure 4b–d shows the TEM images of Bi_2_WO_6_, CuS and 0.1% CuS/Bi_2_WO_6_ composite. The TEM results further validated the nanosheet structures of Bi_2_WO_6_ and the microsphere structure of CuS assembled by nanosheets. The CuS and Bi_2_WO_6_ containing in 0.1% CuS/Bi_2_WO_6_ heterostructure were verified by HRTEM, as shown in Figure 4e. The lattice fringe spacing between adjacent crystal faces is 0.273 nm and 0.189 nm, which can be attributed to the (200) and (110) crystal plane of Bi_2_WO_6_ phase and CuS phase, respectively [40,41], leading to the formation of heterojunction.

### 3.2. Photocatalytic and Photo-Fenton Catalytic Performance

The photo-Fenton and photocatalytic degradation of organic pollutants are shown in Figure 5. As can be seen from Figure 5a, the concentration of initial RhB hardly changes without catalyst, which fully proves the stability of its own molecules. However, after the addition of H_2_O_2_ alone, the degradation rate of RhB can reach 14%. In addition, compared to pure Bi_2_WO_6_, the loading of CuS slightly decreased its photocatalytic activity, which may be due to CuS covering up the active sites, eventually resulting in a decrease in activity.

As can be seen from Figure 5b, RhB was only degraded by 10% when CuS is added as a catalyst alone, which may be due to its high electron–hole recombination rate as a photo-Fenton catalyst. It is important to note that the degradation performance of RhB by photo-Fenton process over CuS/Bi_2_WO_6_ heterostructures is obviously enhanced. The as-prepared 0.1% CuS/Bi_2_WO_6_ heterostructure represents the optimal amount relative to photo-Fenton activity. Figure 5c shows that the rate constant implemented by the equation ln (C_0_/C) = *k_app_*·*t*, where *k_app_*/min^−1^ means the apparent rate constant. The degradation efficiency of RhB (20 mg·L^−1^) can reach nearly 100% within 25 min, the apparent rate constant (*k_app_*/min^−1^) is approximately 40.06 times and 3.87 times higher than that of pure CuS and Bi_2_WO_6_, respectively (Figure 5d). Hence, it can be concluded that loading CuS nanosheets on Bi_2_WO_6_ nanosheets can remarkably promote photo-Fenton catalytic performance.

In order to show that the catalyst can handle a wide variety of wastewaters, the photocatalytic and photo-Fenton catalytic activity of the CuS/Bi_2_WO_6_ heterostructures in TC-HCl aqueous solution were also investigated. Figure 6a shows the variation of TC-HCl concentration of the degradation in different samples. It can be seen the photocatalytic effect of the 0.1% CuS/Bi_2_WO_6_ heterostructure is lower than that of pure Bi_2_WO_6_, which is the same as the activity of the comparison of the dye. At the same time, it is noted in Figure 6b that after the H_2_O_2_ is added, the photo-Fenton catalytic degradation efficiency was close to 73% in 50 min through the constructed 0.1% CuS/Bi_2_WO_6_ photo-Fenton system, which was obviously better than pure Bi_2_WO_6_.

We further explored the effect of H_2_O_2_ concentration, catalyst dosage, and initial pH value on the degradation performance of TC-HCl by 0.1% CuS/Bi_2_WO_6_ heterostructure under visible light irradiation. Firstly, the effect of different H_2_O_2_ concentrations on the degradation of TC-HCl is shown in Figure 7a. As the initial H_2_O_2_ concentration increased from 50 μL to 100 μL, the degradation efficiency increased gradually from 68.1% to 72.8%, while the initial H_2_O_2_ concentration further increased from 100 μL to 200 μL, the degradation efficiency increased slightly. From an economic point of view, the optimal concentration of H_2_O_2_ was chosen as 100 μL. This indicates that adding the appropriate amount of H_2_O_2_ is advantageous for increasing the photo-Fenton catalytic activity. Secondly, the effect of the catalyst dosage on the degradation of TC-HCl is shown in Figure 7b. It can be seen that there is an optimum amount of catalyst and the photo-Fenton activity is highest at this time. When the optimum amount is exceeded, the activity is gradually decreased, which may be because too much catalyst blocks the light, and it eventually leads to a decrease in photo-Fenton catalytic activity. Finally, the effect of pH on the degradation of TC-HCl is shown in Figure 7c. As the pH increased from 5 to 7, the degradation efficiency of TC-HCl increased, and then continued to increase pH to 9, and the degradation activity declined instead, which indicated that it has better photo-Fenton activity under natural pH conditions. This is because under natural conditions, the photo-Fenton reaction is more likely to generate enough active free radicals to degrade pollutants.

The cyclic experiments of TC-HCl were also carried out four times over 0.1% CuS/Bi_2_WO_6_ heterostructure (Figure 8a). It can be seen that the catalyst still maintains a high degradation efficiency after each cycle. Figure 8b presents the XRD patterns of fresh and reused 0.1% CuS/Bi_2_WO_6_ heterostructure in the photo-Fenton reaction. No impurity peaks were presented and only the peak intensity slightly decreased, which suggested that the catalyst has relatively stability. This showed that Cu^2+^ leaching problem in the traditional homogeneous Fenton reaction can be significantly restrained by employing CuS/Bi_2_WO_6_ heterostructures. According to the above results, the 0.1% CuS/Bi_2_WO_6_ heterostructure exhibited significantly enhanced catalytic performance compared with the previously reported catalysts for the degradation of dyes and antibiotics (Table 1), for instance, Bi_2_WO_6_/Fe_3_O_4_ [27], g-C_3_N_4_/CuS [29], TiO_2_/Fe_3_O_4_ [42], etc. This comparison clearly reveals that the 0.1% CuS/Bi_2_WO_6_ heterostructure is a potential candidate for treating dye and antibiotic wastewater under visible light irradiation.

### 3.3. Photocatalytic and Photo-Fenton Catalytic Mechanism

Figure 9 shows the UV−Vis DRS spectra of Bi_2_WO_6_, CuS and 0.1% CuS/Bi_2_WO_6_ heterostructure. Obviously, CuS exhibits a stronger absorption band in visible region, while Bi_2_WO_6_ exhibits weak absorption in visible region. After assembling CuS nanosheets on the surface of Bi_2_WO_6_ nanosheets, the absorption of visible region by the composite catalyst was slightly enhanced. Nevertheless, when light and H_2_O_2_ were simultaneously introduced, the heterogeneous photo-Fenton system formed by CuS and Bi_2_WO_6_ exhibited higher catalytic performance in the case of extremely low Cu^2+^ loading.

The recombination rate of photogenerated electrons and holes of the materials were analyzed by photoluminescence (PL) spectroscopy. In general, the weak fluorescence intensity of the samples indicates that the recombination rate of photogenerated electrons and holes in the material is relatively low. It can be seen from Figure 10a that the Bi_2_WO_6_ has a higher recombination rate of photogenerated electrons and holes than other samples. The intensity of the diffraction peak of the 0.1% CuS/Bi_2_WO_6_ heterostructure is much lower than that of the Bi_2_WO_6_ alone, suggesting that the combination of Bi_2_WO_6_ and CuS can effectively inhibit the recombination of electrons and holes. This is beneficial for the photo-Fenton catalytic reaction involving H_2_O_2_. The time-resolved photoluminescence (TR-PL) spectroscopy of Bi_2_WO_6_ and 0.1% CuS/Bi_2_WO_6_ heterostructure are also used to detect the carrier lifetime, and the results were shown in Figure 10b. The photoelectron lifetimes conform to the two-exponential decay model. Decay times (τ_1_ and τ_2_) and PL aptitudes (A_1_ and A_2_) are given in Table 2. Compared to the Bi_2_WO_6_ (0.903 ns), the photogenerated charge carriers lifetime of 0.1% CuS/Bi_2_WO_6_ heterostructure is significantly prolonged (1.360 ns). The increased carriers lifetime can effectively promote the cycle of Cu^2+^ to Cu^+^, and then Cu^+^ can react with H_2_O_2_ to generate ·OH with strong oxidation performance, thus improving the photo-Fenton catalytic performance.

To investigate the catalytic mechanism, an in situ ESR study was employed to monitor the formation of active radicals. When DMPO (a spin-trapping reagent for·OH and O_2_^−^) was added to the reaction systems, no signal of DMPO-·O_2_^−^ and DMPO-·OH adducts can be detected in the dark (Figure 11), suggesting that negligible OH and·O_2_^−^ radical concentrations were present in the solution. However, under the visible light irradiation, both pristine Bi_2_WO_6_ and CuS/Bi_2_WO_6_ heterostructure can produce ESR signals (Figure 11). The characteristic signals of DMPO-·OH with a four-line ESR signal, relative intensities of 1:2:2:1, and α_H_ = α_N_ = 1.5 mT (Figure 11a) were observed [43,44]. Similarly, a doublet of triplets ESR signal with relative intensities of 1:1:1:1 characteristic signals of DMPO-·O_2_^−^ (Figure 11b) was also observed [45,46]. However, it is worth noting that the coupling constants of DMPO-·O_2_^-^ adduct are a_H_ = 0.92 mT, a_N_ =1.4 mT, respectively. The value of a_H_ does not match with the normal value, which may be ascribed to the effects of the solution environment such as pH and solvent. According to the literature reported and our experimental result, the doublet of triplets ESR signal was assigned to DMPO-·O_2_^−^ adduct rather than those centerd on hetero atoms such as O or S [47,48]. Furthermore, compared to pristine Bi_2_WO_6_, the highly intensive signals were observed for·OH and·O_2_^−^ adducts in the 0.1% CuS/Bi_2_WO_6_ heterostructure, indicating that CuS/Bi_2_WO_6_ heterostructure can generate abundant ·OH and ·O_2_^−^ in the presence of visible light and H_2_O_2_, which is conducive to the degradation of pollutants.

The band structure of CuS/Bi_2_WO_6_ heterostructure was studied by Tauc plots and XPS valence band spectrum analysis. The measured band gap diagram shows that Bi_2_WO_6_ and CuS, correspond to band gap energies of 2.65 eV and 1.54 eV according to the equation of Kubelka–Munk function, respectively [49], as shown in Figure 12a,b. In addition, according to the intercept with X axis obtained from XPS results, the position of valence band of the material is further determined, indicating that valence band (VB) of Bi_2_WO_6_ and CuS are 2.24 and 1.46 V, respectively (Figure 12c,d). Therefore, following the empirical equation: E_CB_ = E_VB_ − E_g_, the conduction band minimum (CBM) of Bi_2_WO_6_ and CuS are confirmed to be −0.41 and −0.08 V, respectively.

In order to further investigate the role of CuS/Bi_2_WO_6_ heterostructures in photo-Fenton catalysis system, the enhancement mechanism of photo-Fenton activity is discussed, and the schematic diagram is shown in Figure 13a,b. Due to the n-type nature of Bi_2_WO_6_ and p-type nature of CuS, the Fermi levels of Bi_2_WO_6_ and CuS are located nearer to the CBM and VBM, respectively. Upon contact between Bi_2_WO_6_ and CuS, due to Fermi level equilibrium the energy level of CuS is upshifted and Bi_2_WO_6_ is downshifted leading to the formation of p-n heterojunction [41]. The p-n junction also led to the generation of an internal electric field where the CuS region is negatively charged and the Bi_2_WO_6_ region is positively charged. This internal field is responsible for efficient separation of the charge carriers. When irradiated with light, both Bi_2_WO_6_ and CuS are capable of absorbing visible light photons to produce e^−^/h^+^ pairs. Due to favorable band alignment, the photogenerated electrons can migrate from the CB of CuS to Bi_2_WO_6_, whereas the migration of holes can take place in reverse direction in the VB of both semiconductor components. This cyclic movement of photogenerated carriers is characteristic of a type-II p-n heterojunction, which leads to their efficient space separation. In the heterogeneous visible light photo-Fenton system constructed in this experiment, the active species involved in the oxidative degradation of organic pollutants mainly include photogenerated holes, O_2_^−^ and ·OH radical.

When visible light irradiated on the surface of the CuS/Bi_2_WO_6_ heterostructure, electrons on the Bi_2_WO_6_ valence band were excited to the conduction band (Equation (1)). Both in the process of degrading RhB and TC-HCl, some electrons in the CB of Bi_2_WO_6_ could reduce O_2_ into ·O_2_^−^ because the CB potential of Bi_2_WO_6_ (−0.41 eV vs. NHE) was more negative than O_2_/·O_2_^−^ potential (−0.33 or −0.046 eV vs. NHE) (Figure 13a,b; Equation (2)). The other electrons could transfer to the surface of CuS to react with H_2_O_2_ to generate ·OH via the photo-Fenton reaction (Equations (4)–(7)), while the photogenerated holes left on the valence band of Bi_2_WO_6_ could directly degrade RhB and TC-HCl or react with OH^−^ to generate ·OH (Equation (3)). During the degrading of RhB, the VB holes of Bi_2_WO_6_ could oxidize OH^−^ into OH because the VB potential of Bi_2_WO_6_ (2.24 eV vs. NHE) was more positive than that of ·OH/OH^−^ potential (1.99 eV vs. NHE) (Figure 13a), but could not oxidize OH^-^ into ·OH during the degradation of TC-HCl because the VB potential of Bi_2_WO_6_ (2.24 eV vs. NHE) was more negative than ·OH/OH^-^ potential (2.38 eV vs. NHE) (Figure 13b). This resulted in a heterogeneous photo-Fenton process to degrade the organic pollutants via the generated active species (Equation (8)). Possible reactions are described below:

Photocatalysis:CuS(Cu^II^)/Bi_2_WO_6_→CuS/Bi_2_WO_6_(h^+^ + e^−^)(1)
H_2_O_2_+CuS/Bi_2_WO_6_(e^−^)→·OH O_2_+e^-^→·O_2_^−^(2)
CuS/Bi_2_WO_6_(h^+^) + OH^-^→·OH(3)

Fenton reaction:≡Cu^II^S/Bi_2_WO_6_ + H_2_O_2_ → H_2_O_2_≡Cu^II^S/Bi_2_WO_6_(4)
H_2_O_2_≡Cu^II^S/Bi_2_WO_6_ → ≡Cu^I^S/Bi_2_WO_6_ + HO_2_ + H^+^(5)
≡Cu^II^S/Bi_2_WO_6_ +·HO_2_ → ≡Cu^I^S/Bi_2_WO_6_ + O_2_ + H^+^(6)
≡Cu^I^S/Bi_2_WO_6_ + H_2_O_2_ → ≡Cu^II^S/Bi_2_WO_6_ + ·OH + OH^-^(7)
Organic pollutants +·O_2_^−^/·OH/h^+^ → … → CO_2_ + H_2_O + …(8)

## 4. Conclusions

In summary, we have developed a simple in situ assembly strategy to construct the two-dimensional (2D) in-plane CuS/Bi_2_WO_6_ sheet-on-sheet p-n heterostructures and used as photo-Fenton catalysts. A series of techniques were employed to characterize the physical and chemical properties of as-synthesized CuS/Bi_2_WO_6_ heterostructures. Photo-Fenton catalytic treatment of organic pollutants was successfully achieved by employing CuS/Bi_2_WO_6_ heterostructures as photocatalysts in the presence of H_2_O_2_. Photo-Fenton catalytic activity and stability can be significantly promoted by assembling CuS nanosheets on Bi_2_WO_6_ nanosheets. The excellent photo-Fenton catalytic performance can be attributed to the high-efficiency charge separation and migration of CuS/Bi_2_WO_6_ p-n heterostructures. The photo-Fenton reaction of CuS on the surface of the heterogeneous catalyst with H_2_O_2_ produces a large number of·O_2_^−^ and ·OH, which are responsible for the degradation of the organic pollutants. The effects of different factors such as H_2_O_2_ concentration, catalyst dosage, and initial pH value on TC-HCl degradation were systematically investigated, and the enhanced catalytic oxidation mechanism of 2D/2D CuS/Bi_2_WO_6_ p-n heterogeneous photo-Fenton system was also elucidated. The present work offers a simple strategy to construct 2D in-plane heterostructures with efficient photo-generated electron–hole separation and high stability, and can be used for wastewater purification via photo-Fenton reaction.

## Figures and Tables

**Figure 1 nanomaterials-09-01151-f001:**
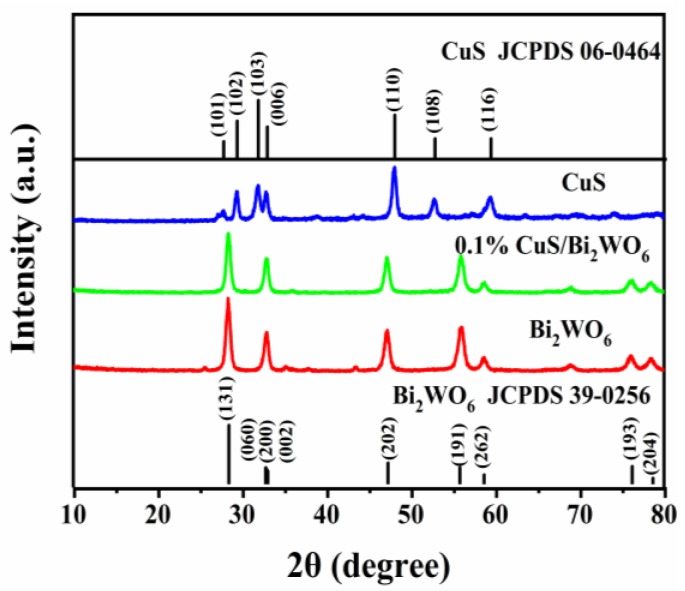
XRD patterns of the CuS, Bi_2_WO_6_ nanosheets and 0.1% CuS/Bi_2_WO_6_ heterostructure.

**Figure 2 nanomaterials-09-01151-f002:**
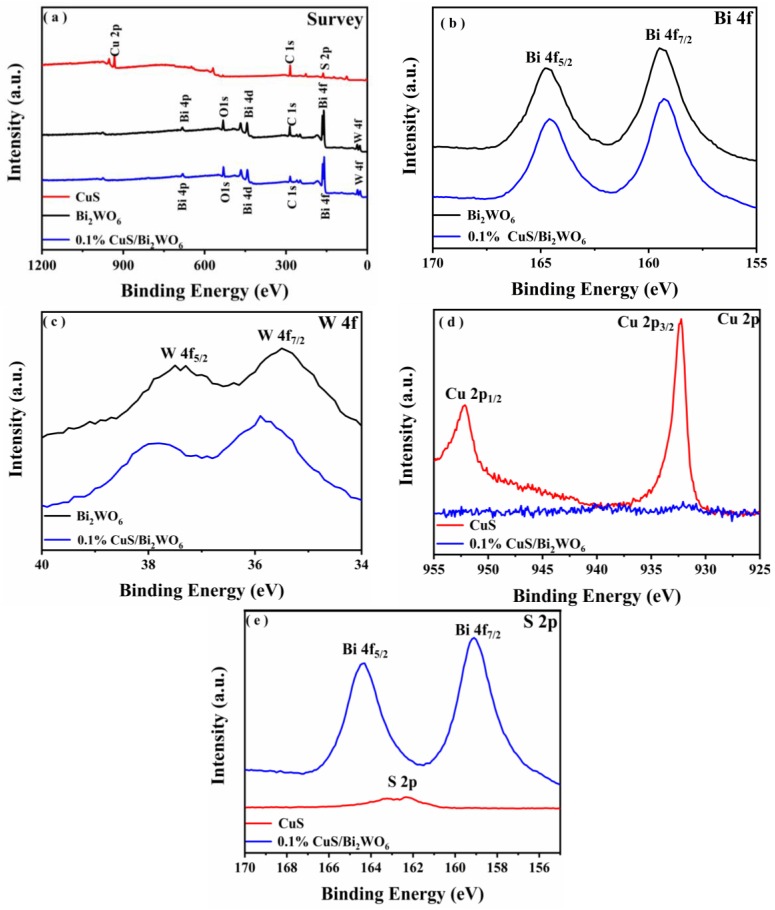
(**a**) The XPS survey spectra of Bi_2_WO_6_, CuS, and 0.1% CuS/Bi_2_WO_6_ heterostructure (**b**–**e**) High resolution XPS spectra of the Bi_2_WO_6_, CuS, and 0.1% CuS/Bi_2_WO_6_ heterostructure.

**Figure 3 nanomaterials-09-01151-f003:**
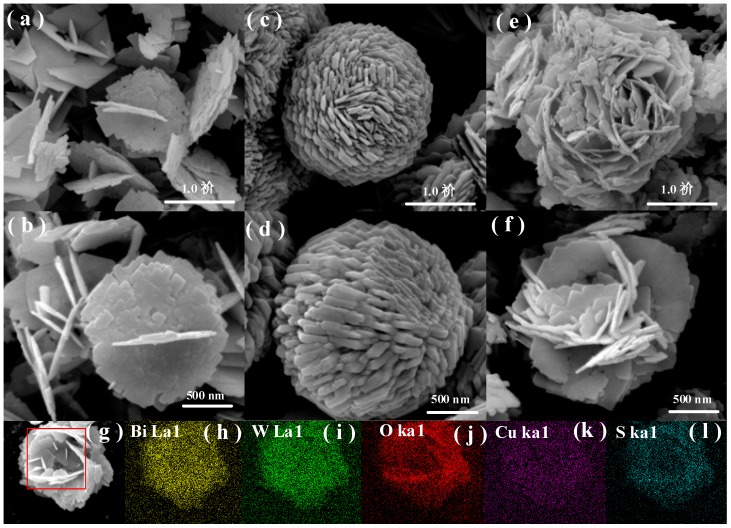
FESEM images of (**a**,**b**) Bi_2_WO_6_, (**c**,**d**) CuS, and (**e**,**f**) 0.1% CuS/Bi_2_WO_6_ heterostructure, and energy filtered elemental mapping of 0.1% CuS/Bi_2_WO_6_ heterostructure (**g**–**l**).

**Figure 4 nanomaterials-09-01151-f004:**
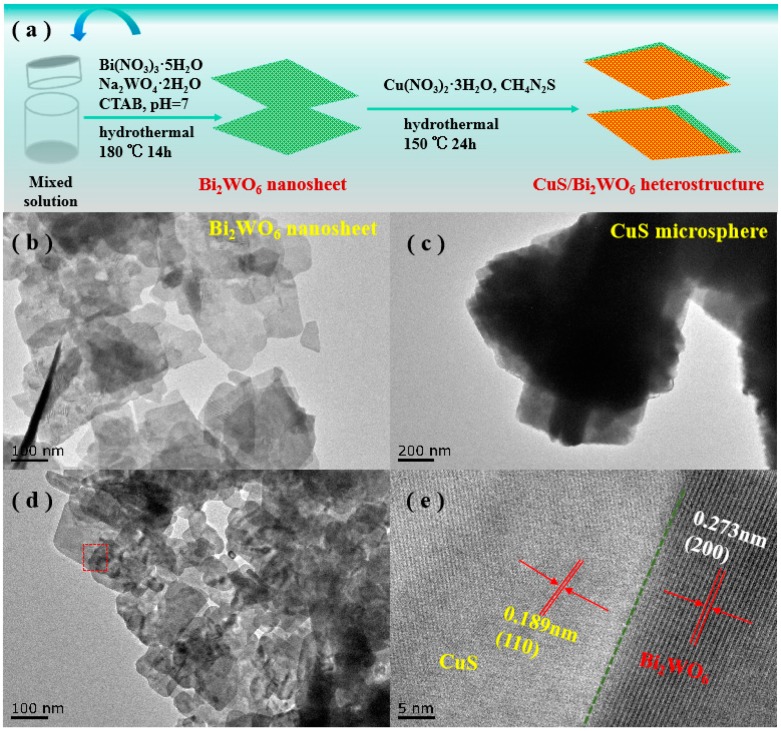
(**a**) Schematic preparation of CuS/Bi_2_WO_6_ sheet-onto-sheet p-n heterostructures. (**b**) TEM images of Bi_2_WO_6_. (**c**) TEM images of CuS. (**d**) TEM and (**e**) HRTEM image of as-prepared 2D/2D 0.1% CuS/Bi_2_WO_6_ sheet-onto-sheet heterostructure.

**Figure 5 nanomaterials-09-01151-f005:**
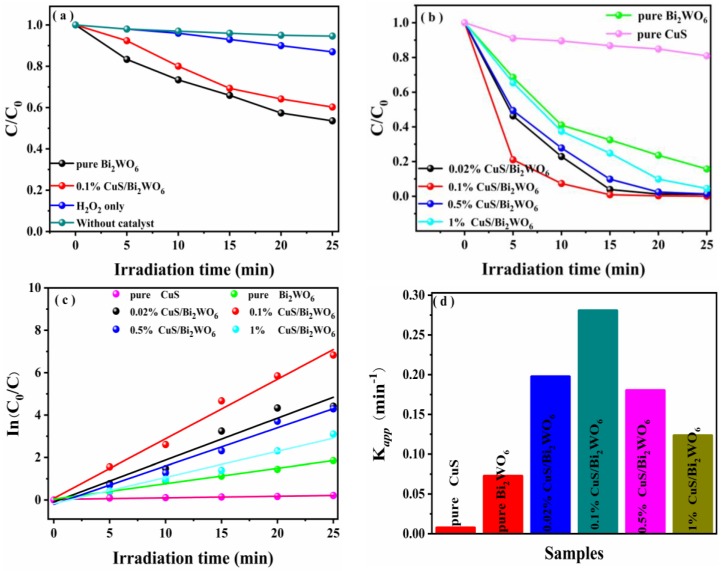
(**a**) The photo-catalytic degradation dynamic curves of RhB in the presence of pure Bi_2_WO_6_, H_2_O_2_ only, without catalyst, and 0.1% CuS/Bi_2_WO_6_ heterostructure. (Reaction conditions: T = 25 °C, concentration of RhB = 20 mg·L^−1^, and catalyst concentration = 1 g·L^−1^). (**b**) The photo-Fenton catalytic degradation dynamic curves of RhB in the presence of pure Bi_2_WO_6_, CuS, and a series of CuS/Bi_2_WO_6_ heterostructures. (Reaction conditions: T = 25 °C, the dosage of H_2_O_2_ = 100 μL, concentration of RhB = 20 mg·L^−1^, and catalyst concentration = 1 g·L^−1^). (**c**) The pseudo-first-order reaction kinetics and (**d**) values of reaction rate constants over the synthesized samples in the photo-Fenton reaction.

**Figure 6 nanomaterials-09-01151-f006:**
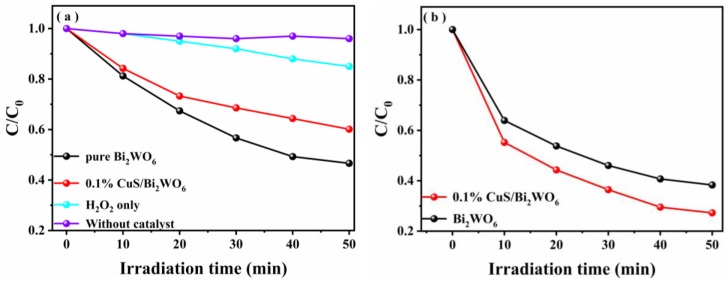
(**a**) The photo-catalytic degradation dynamic curves of TC-HCl in the presence of pure Bi_2_WO_6_, H_2_O_2_ only, without catalyst, and 0.1% CuS/Bi_2_WO_6_ heterostructure. (Reaction conditions: T = 25 °C, concentration of TC-HCl = 40 mg·L^−1^, and catalyst concentration = 1 g·L^−1^). (**b**) The photo-Fenton catalytic degradation dynamic curves of TC-HCl in the presence of pure Bi_2_WO_6_ and 0.1% CuS/Bi_2_WO_6_ heterostructure (Reaction conditions: T = 25 °C, the dosage of H_2_O_2_ = 100 μL, concentration of TC-HCl = 40 mg·L^−1^, and catalyst concentration = 1 g·L^−1^).

**Figure 7 nanomaterials-09-01151-f007:**
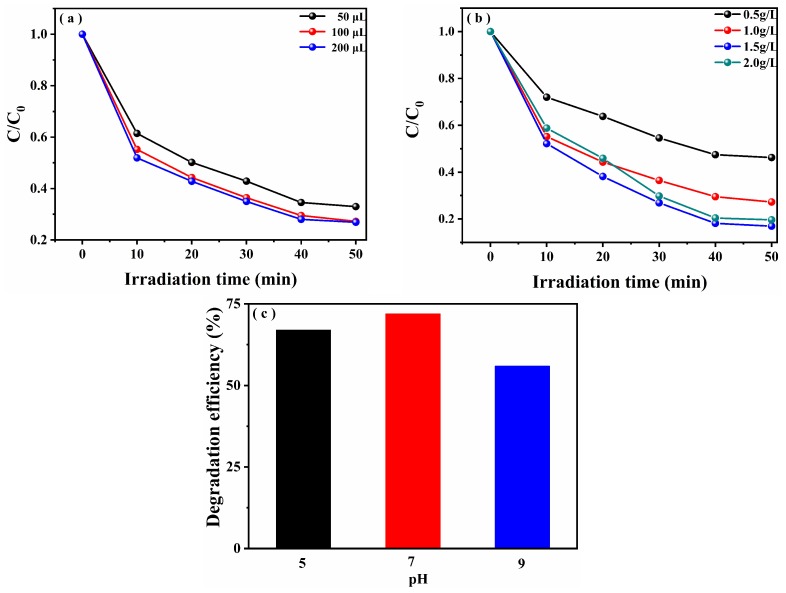
Effects of H_2_O_2_ concentration (**a**), catalyst dosage (**b**) and initial pH value (**c**) on the photo-Fenton catalytic activity for TC-HCl removal.

**Figure 8 nanomaterials-09-01151-f008:**
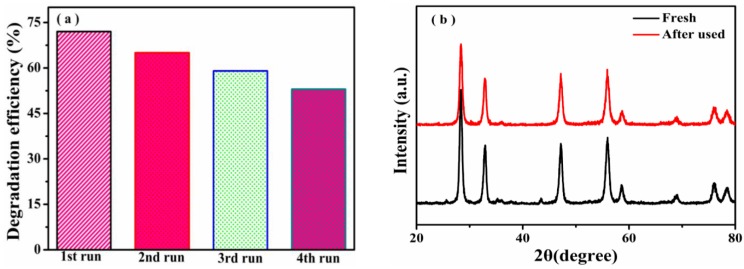
(**a**) Degradation efficiency of TC-HCl during different cycles, (Reaction conditions: initial pH value = 7, T = 25 °C, initial H_2_O_2_ concentration = 100 μL, concentration of TC-HCl = 40 mg·L^−1^, and catalyst dosage = 1 g·L^−1^); (**b**) XRD patterns of 0.1% CuS/Bi_2_WO_6_ heterostructure before and after photo-Fenton reaction.

**Figure 9 nanomaterials-09-01151-f009:**
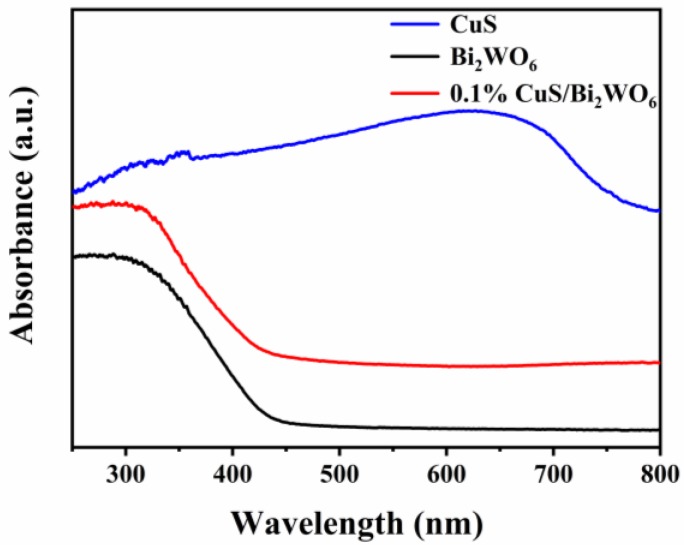
UV-Vis DRS spectrum of Bi_2_WO_6_, CuS, and 0.1% CuS/Bi_2_WO_6_ heterostructure.

**Figure 10 nanomaterials-09-01151-f010:**
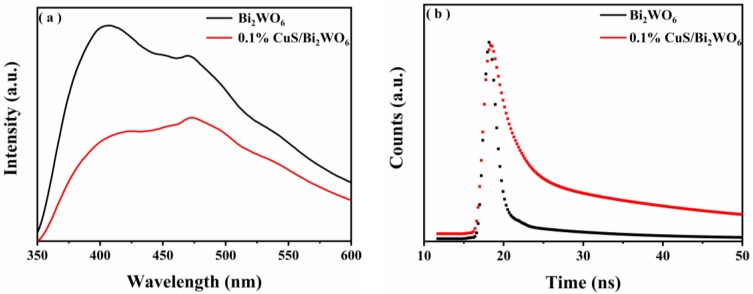
Photoluminescence (PL) spectra of Bi_2_WO_6_ and 0.1% CuS/Bi_2_WO_6_ heterostructure (**a**), time-resolved transient PL decay curves of Bi_2_WO_6_ and 0.1% CuS/Bi_2_WO_6_ heterostructure (**b**).

**Figure 11 nanomaterials-09-01151-f011:**
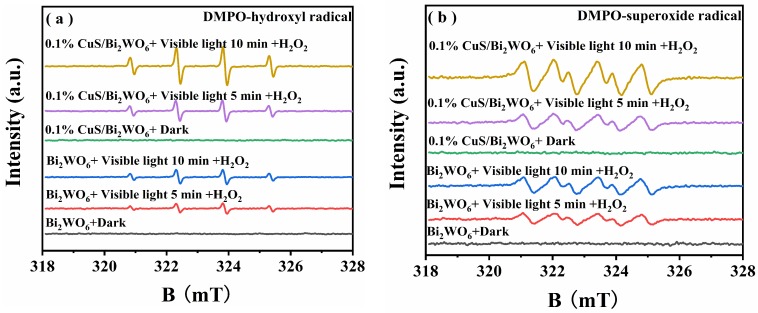
DMPO spin-trapping ESR spectra of Bi_2_WO_6_ and 0.1% CuS/Bi_2_WO_6_ heterostructure for (**a**) DMPO-·OH in aqueous dispersion and (**b**) DMPO-·O_2_^−^ in methanol dispersion in the dark or under visible light irradiation and in the presence of H_2_O_2_, respectively.

**Figure 12 nanomaterials-09-01151-f012:**
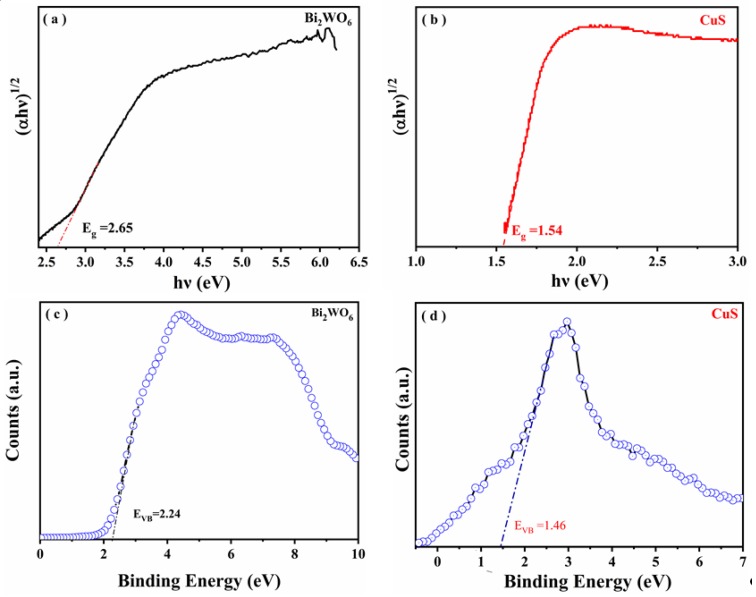
Tauc plots of (αhν)^1/2^ vs. hν for the samples Bi_2_WO_6_ (**a**) and CuS (**b**) and XPS valence band spectra of Bi_2_WO_6_ (**c**) and CuS (**d**).

**Figure 13 nanomaterials-09-01151-f013:**
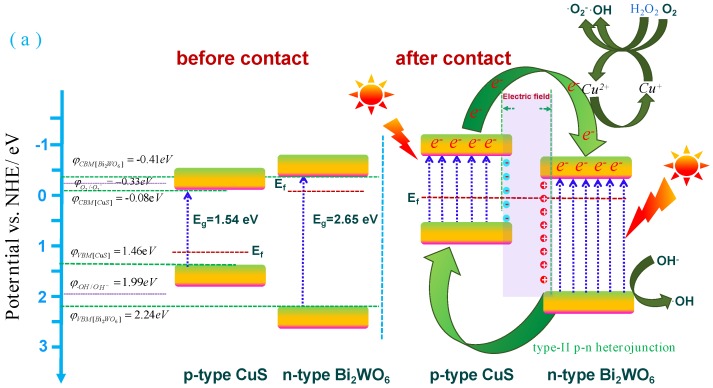
(**a**) The possible photocatalytic and photo-Fenton catalytic mechanism of CuS/Bi_2_WO_6_ heterostructures for degradation of RhB and (**b**) TC-HCl under visible light irradiation.

**Table 1 nanomaterials-09-01151-t001:** Comparison of photo and photo-Fenton catalytic performance for removal of pollutants.

Samples	Light Source		Organic Pollutants			Experimental Conditions			η^d)^ (%)	Refs.
	Type ^a)^	Power (W)	Type	Conc. ^b)^ (mg·L^−1^)	Catalyst (g/L)	t ^c)^ (time)	pH	H_2_O_2_		
Co_3_O_4_/Bi_2_WO_6_	Tungsten lamp	100	Methylene blue (MB)	10	1	80	8	1.96 mM	96	[26]
Bi_2_WO_6_/Fe_3_O_4_	Xe lamp	350	RhB	10	0.5	120	7	10 mM	95	[27]
CuS-TiO_2_	Xe lamp	350	MB	10	1	180	-	-	100	[28]
g-C_3_N_4_/CuS	Xe lamp	300	RhB	10	0.3	60	-	-	96.8	[29]
CuS/Bi_2_W_2_O_9_	Xe lamp	150	Diuron	10	0.75	180	-	200 µL	95	[31]
CuS/Bi_4_Ti_3_O_12_	Xe lamp	250	2-methyl-4-chlorophenoxyacetic acid	10	0.25	180	-	150 µL	96	[41]
TiO_2_/Fe_3_O_4_	UVC-lamp	10	TC-HCl	50	0.3	60	7	10 mM	98	[42]
0.1% CuS/Bi_2_WO_6_	Metal halide lamp	300	RhB	20	1.0	25	-	100 µL	100	this work
			TC-HCl	40	1.0	50	7		73	

^a)^ UVC-lamp were the UV-light sources, Xe lamp, tungsten lamp and metal halide lamp were the visible-light source; ^b)^ concentration of pollutants; ^c)^ irradiation time; ^d)^ the total removal efficiency of organic pollutant.

**Table 2 nanomaterials-09-01151-t002:** Parameter of time-resolved PL decay curves.

Samples	τ_1_ (ns)	A_1_	τ_2_ (ns)	A_2_	Τ_av_ (ns)
Bi_2_WO_6_ nanosheets	0.903	2.22 × 10^11^	12.205	1022.997	0.903
0.1% CuS/Bi_2_WO_6_	17.552	1.50 × 10^3^	1.359	1.32 × 10^9^	1.360

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
