# Peer review of "2D In-Plane CuS/Bi2WO6 p-n Heterostructures with Promoted Visible-Light-Driven Photo-Fenton Degradation Performance"

_nanomaterials, 2019, doi:10.3390/nano9081151_

Round 1

Reviewer 1 Report

Comments (2) after rewrite.

There are still some problems with the ESR sections of the paper,

Line 132 the instrument is a Bruker ELEXSYS-II 500, not LEXSYS as written

Line 141 I’m sure you mean a ‘quartz ESR tube’, not a ‘quartz nuclear magnetic tube’ as written

Line 339 the relative intensities should be 1:1:1:1:1:1 and not 1:1:1:1 as written.

Are you sure that the spectra in 11a and 11b are recorded under exactly similar conditions ? I find the increase in S/N remarkable as even in similar MeOH reaction media, the S/N of the superoxide-DMPO adduct is always less than for the hydroxyl adduct – see references 44, 45.

The authors of this paper publish the kind of spectra I expect to see in….. Catal. Sci. Technol., 2019, 9, 3193-3202, which also seems to have the coupling constant values closer to those expected for the superoxide adduct. If you present the data in the current submission in the same format, it would be less confusing. Present the ESR data here as scans of 100G so that all the spectrum is fitted in to the diagram.

Captions on Figure 13 need correcting (Befeore, ?? Before surely)

In Table 1 you introduce a range of other potential pollutants, from previously published work,  rather than merely Rhodamine B and Tetracycline hydrochloride, there is no discussion of this additional data, and the pollutant MB is not defined. I presume it is methylene blue, but maybe I am wrong. Please tighten this up.

Author Response

Dear reviewer,

Re: Submission of a revised manuscript about nanomaterials-550057 to Nanomaterials

We thank you for your review regarding the our manuscript nanomaterials-550057 entitled “2D in-Plane CuS/Bi2WO6 p-n Heterostructures with Promoted Visible-light-driven Photo-Fenton degradation performance”. Based on yours and your comments, we have carefully revised this manuscript and responded to the comments point-by-point. A revised version and the response letter are attached, in which the revised parts are highlighted in yellow background.

Herein we greatly thank to you for your constructive suggestions and important comment on this manuscript.

Look forward to hearing from you soon.

Yours sincerely,

Prof. Dr. Danjun Wang

Reviewer 2 Report

I am satisfied what the authors made for the improvement of their paper. Thus, in my view, it can be accepted for publication as it stands now.

Author Response

Dear reviewer,

Re: Submission of a revised manuscript about nanomaterials-550057 to Nanomaterials

We thank you for your review regarding the our manuscript nanomaterials-550057 entitled “2D in-Plane CuS/Bi2WO6 p-n Heterostructures with Promoted Visible-light-driven Photo-Fenton degradation performance”. Based on yours and your comments, we have carefully revised this manuscript and responded to the comments point-by-point. A revised version and the response letter are attached, in which the revised parts are highlighted in yellow background.

Herein we greatly thank to you for your constructive suggestions and important comment on this manuscript.

Look forward to hearing from you soon.

Yours sincerely,

        Dr. Danjun Wang

Round 2

Reviewer 1 Report

This revised version is an enormous improvement. The ESR data is presented in a much clearer manner, well done.

My only remaining comment concerns the referencing. You must check through all the references please. I found a serious mistake in one selected at random.

Reference 33 actually appears in Nature Communications and not Nature Chemistry as written, and the reference should read Nat. Comm. 2015, 6 : article 8340

There is nothing more irritating in research than to find mis-quoted references. It took quite a while for me to track down the real reference 33.... wrong journal is an epic fail !!

In the title of reference 29, p-n must be italicised as p-n 

Because I have found 2 errors, there may well be more - please check them all by viewing the title page of each reference to make sure all the details are correct

Providing you promise to do this, I do not need to see the manuscript again.

Author Response

Dear reviewer,

Re: Submission of a revised manuscript about nanomaterials-550057 to Nanomaterials

We thank you for your review regarding the our manuscript nanomaterials-550057 entitled “2D in-Plane CuS/Bi2WO6 p-n Heterostructures with Promoted Visible-light-driven Photo-Fenton degradation performance”. Based on yours and your comments, we have carefully revised this manuscript and responded to the comments point-by-point. A revised version and the response letter are attached, in which the revised parts are highlighted in yellow background.

Herein we greatly thank to you for your constructive suggestions and important comment on this manuscript.

Look forward to hearing from you soon.

Yours sincerely,

Prof. Dr. Danjun Wang

This manuscript is a resubmission of an earlier submission. The following is a list of the peer review reports and author responses from that submission.

Round 1

Reviewer 1 Report

It is an important paper, a good report about a professional level research. The heterostructure is very well characterized and overall the work merits publication. However, more than slight linguistic improvements are required. For instance, I have no idea what the first sentence in the Introduction means. I think that there is room for a thorough linguistic brush-up by a native professional.

Author Response

Response: Thanks for the your suggestion. Based on yours comments, we have thoroughly checked and revised the manuscript by a native speaker.

Reviewer 2 Report

While I think the objectives of this paper are fine – good, new photocatalysts are an essential part of our attack on pollutants from a variety of sources, there are indications that this paper lacks the necessary experimental rigour in one particular area.

DMPO Spin trapping by ESR is not an easy experiment to get right, and I have doubts about the identity attributed to the spin adduct in Figure 11b, from the coupling constants derived from the spectrum, there is little to suggest that this spectrum arises from the superoxide spin adduct of DMPO.

That this particular team is inexperienced in ESR techniques is witnessed by the sentence “The electron spin resonance (ESR) spectra were obtained on a Bruker model JES-FA200 spectrometer…..”. This spectrometer is actually made by the company JEOL and not Bruker.

None of the critical experimental details are given. What was the concentration of DMPO? How was the solution prepared to remove artefacts prior to measurement? Was the experiment sacrificial (in a closed tube/cell) or were aliquots drawn from a bulk reactor and spun down with the supernatant then measured. What was the initial pH of the system? What were the relative kinetics of adduct formation? What was the modulation amplitude and time constant of the spectrometer ? Were the spectra 1K, 2K or 4K points? What were the decay kinetics of the spectra once the light source (lamp or laser) was extinguished? These are all critical points in determination of the adduct formed in 11b, as my following comments demonstrate.

Trapping of the hydroxyl radical by DMPO is fast, around 3.5 x 109.M-1.sec-1, approaching diffusion controlled rates. On the other hand, trapping of the other potential candidates involved in photocatalysis, hydroperoxyl, HOO and superoxide O2-• are 6 to 8 times slower, at 6.6 x 103 M-1.sec-1 for hydroperoxyl and 1.0 to 1.2 M-1.sec-1 depending on pH. Moreover, decay rates for the adduct radicals, under broadly similar experimental conditions are also very different. The half life for the decay of the DMPO-OH adduct, can be as long as 14h, while the half life of the DMPO-OOH adduct is » 80sec, and that of the DMPO-OO-• adduct is 50-60sec. These data indicate that under broadly similar conditions the S/N for the DMPO-OH adduct in 11a should be far greater than that for the superoxide adduct. In Figure 11, unfortunately, the S/N for 11b is far greater than 11a, which means it is highly unlikely that this adduct is from superoxide.

In addition, the coupling constants do not quite match up. In 11a the spectrum is quite normal aN =aH »14G, however with a ruler and 11b, aN » 13G while AH » 8G. This does not match with the normally accepted value for aH which should be between 11-12G. Other possible candidates for the identity of the radical adding to DMPO could be those centred on hetero atoms such as O or S. Thes would also show the general property of aH<aN.

I urge the authors to perform more background experiments and consider the comments above before rewriting and re-submitting the manuscript. I urge you to do this, the work is important.

Author Response

Response: Thanks for your constructive comments, which help us greatly improve the quality of the manuscript, and we have made relevant revisions thoroughly according to these comments. Based on yours comments, we have carefully revised this manuscript and responded to your comments point-by-point. A revised version and the response letter are attached, in which the revised parts are highlighted in yellow background.

Reviewer 3 Report

2D in-Plane CuS/Bi2WO6 p-n Heterostructures with Promoted Visible-Light-Driven Photo-Fenton Degradation Performance

The manuscript studied the employ 2D in-Plane CuS/Bi2WO6 p-n Heterostructures for the photo-fenton degradation of organic molecules present in aqueous solution. The manuscript is rich of data about both characterization and catalytic activity. The topic is interesting but there are some aspects that the authors should improve.

The manuscript could be published after major revision.

Abstract:

The abstract should be      revised; some characterization results should be added

Please substitute      acronyms such as TC-HCl and RhB

 Introduction:

1.      Among them, industrial wastewater accounts for a large proportion of water pollution”. Please add a references.

Moreover, I agree with  the authors that the water pollution  connected with organic dye in large portion is due to the industrial use, but i disagree about the industrial pollution due to antibiotics. The first cause of  wastewater contaminations due pharmaceuticals  is connected to the use of pharmaceuticals in everyday human life. Generally pharmaceuticals such as antibiotics, are not completely metabolized by the human organism and therefore, eliminated by urine and faeces and then transported into the wastewater treatment plants. The most common worldwide used WWTPs are mainly based on the activated sludge technique and are not designed to treat water polluted with pharmaceuticals even if present at trace levels. Some references are reported below:

N.H. Tran, M. Reinhard, K.Y.-H. Gin, Occurrence and fate of emerging contaminants in municipal wastewater treatment plants from different geographical regions-a review, Water Research, 133 (2018) 182-207.

J.H.O.S. Pereira, V.J.P. Vilar, M.T. Borges, O. González, S. Esplugas, R.A.R. Boaventura, Photocatalytic degradation of oxytetracycline using TiO2 under natural and simulated solar radiation, Solar Energy, 85 (2011) 2732-2740.

V.L. Cunningham, S.P. Binks, M.J. Olson, Human health risk assessment from the presence of human pharmaceuticals in the aquatic environment, Regulatory Toxicology and Pharmacology, 53 (2009) 39-45.

2.      ….CuS has been extensively studied in combination with other photocatalysts” please specify the sentences, giving more details about the photocatalysts based on combination CuS and other semiconductors.

3.      Line 57 ·OH instead OH

4.      Line 62, what authors mean for “ultra-small quantity”? Please clarify

5.      Lines 65-72 seems conclusions, please modify in order to  underline the novelty of the paper

Experimental:

It      is important in this stage add the reactor configuration because the catalytic      performances could be influenced by geometrical characteristics of the reactor      and employed during the experimental test. In addition, how the authors measured      the temperature of aqueous solution during the tests?

Results and discussion

Why the authors reports only the      characterization of the sample 0.1%      CuS/Bi2WO?      Please clarify

The discussion of catalytic activity is      confused and the authors do not provide a comparison with the data      reported in literature, please rewrite in order to clarify the obtained      results.

Author Response

Response: Thanks for the referee’s constructive comments, which help us greatly improve the quality of the manuscript, and we have made relevant revisions thoroughly according to these comments. Based on yours comments, we have carefully revised this manuscript and responded to your comments point-by-point. A revised version and the response letter are attached, in which the revised parts are highlighted in yellow background.

Round 2

Reviewer 2 Report

Unfortunately I think there are still improvements to be made to this manuscript. Maybe even more experiments are needed. Your response still shows deficiencies in the ESR work. For example... when you say...

Similarly, a six-line ESRsignal was recorded for ·O2- generation (Fig. 11b) with coupling constants about 8G, which is consistent with the previous literature reported[43-46]. ”(

The spectrum is in fact a doublet of triplets, with a Nitrogen coupling constant and a proton coupling constant. You must measure these correctly and state them. About 8G is not good enough.

Why don't the spectra in 11b overlay exactly, if they are the same species? I just think this presentation can be improved.

Why are you working with small volume sacrificial samples? I think you should be performing the ESR experiments in bulk solutions, away from the spectrometer, then taking small aliquots as time progresses and analysing these.